# Investigation of the Storage and Stability as Well as the Dissolution Rate of Novel Ilaprazole/Xylitol Cocrystal

**DOI:** 10.3390/pharmaceutics16010122

**Published:** 2024-01-17

**Authors:** Sihyun Nam, Changjin Lim, Yongdae Kim, Bokyoung Yoon, Taewoo Park, Woo-Sik Kim, Ji-Hun An

**Affiliations:** 1Functional Crystallization Center, Department of Chemical Engineering, Kyung Hee University, Yongin 17104, Republic of Korea; skatlgus1@khu.ac.kr; 2School of Pharmacy, Jeonbuk National University, Jeonju 54896, Republic of Korea; limcj@jbnu.ac.kr; 3R&D Center, UniCel Lab, Uiwang 16079, Republic of Korea; mac8383@unicellab.com; 4R&D Center GUJU Pharm, Suwon 16229, Republic of Korea; byyoon@guju.co.kr (B.Y.); twpark@guju.co.kr (T.P.)

**Keywords:** cocrystal, ilaprazole, active pharmaceutical ingredient, co-crystallization, stability, dissolution rate

## Abstract

Reflux esophagitis, a treatment for gastric ulcers known as Ilaprazole (Ila), is not stable during storage and handling at room temperature, requiring storage at 5 degrees Celsius. In this study, to address these issues with Ila, coformers rich in oxygen (O) and hydroxyl (OH) groups capable of forming hydrogen bonds with were selected. These coformers included Xylitol (Xyl), Meglumine (Meg), Nicotinic acid (Nic), L-Aspartic acid (Asp), and L-Glutamic acid (Glu). A 1:1 physical mixture of Ila and each coformer was prepared, and the potential for cocrystal formation was predicted using differential scanning calorimetry (DSC) screening. The results indicated the potential for cocrystal formation in the Ila/Xyl physical mixture. Subsequently, Ila and Xyl were mixed in ethyl acetate (EA) in a 1:1 ratio, and after 28 h of slurry, the formation of Ila/Xyl cocrystal was confirmed through solid-state CP/MAS ^13^C NMR spectrum analysis, showing intermolecular hydrogen bonding and conformational changes. Furthermore, the 1:1 ratio of Ila/Xyl cocrystal was confirmed through solution-state NMR (^1^H, ^13^C, and 2D) molecular structure analysis. To assess the stability of Ila/Xyl cocrystal at room temperature, it was stored and compared with Ila at 25 ± 2 °C and relative humidity (RH) of 65 ± 5% over three months. The results showed that the purity of Ila/Xyl cocrystal remained at 99.8% from the initial purity of 99.75% over the three months, while Ila was predicted to decrease from an initial 99.8% purity to 90% after three months. Additionally, at 25 ± 2 °C and RH 65 ± 5%, a specific impurity B in Ila/Xyl cocrystal was observed to be 0.03% over three months, whereas Ila was predicted to increase from an initial 0.032% to 2.28% after three months. To evaluate the dissolution rate of Ila/Xyl cocrystal, a formulation was prepared and compared with Ila at pH 10, with a dosage equivalent to 10 mg of Ila. The results showed that Ila/Xyl cocrystal reached 55% within 15 min and 100% within 45 min, while Ila was predicted to reach 32% at 15 min and 100% only after 60 min. However, overall, the Ila/Xyl cocrystal showed results equivalent to or exceeding the dissolution rate of Ila. Therefore, it is predicted that the Ila/Xyl cocrystal will maximize its effectiveness as a more convenient crystal structure for formulation development, allowing storage and preservation at room temperature without the need for the problematic 5 °C refrigeration during ambient conditions and storage, addressing the issues associated with Ila.

## 1. Introduction

Cocrystal refers to a form where two or more molecules form a specific stoichiometric ratio and crystal structure within a single lattice. In the context of pharmaceuticals, during the formation of cocrystals, the molecule responsible for the crystal structure, possessing activity, is referred to as the API (Active Pharmaceutical Ingredient), while the molecule without activity is termed the coformer. Cocrystals form nonionic or noncovalent bonds, such as van der Waals forces or hydrogen bonding, within or between molecules without the need to break or induce chemical covalent bonds. Depending on the cocrystal formation, the physicochemical properties of the substance, such as solubility, dissolution rate, bioavailability, physical and chemical stability, melting point, density, etc., can change. This alteration in physicochemical characteristics offers a crystal structure that can enhance the therapeutic efficacy of the API [1,2,3,4,5].

Some APIs face challenges such as low solubility, resulting in decreased bioavailability and stability issues. To overcome these difficulties, several studies are employing co-crystallization. In the case of felodipine and xylitol, for instance, when co-crystallized in a 1:2 stoichiometric ratio, the dissolution rate increased up to 79.8% compared to felodipine alone [6]. Moreover, carbamazepine demonstrated enhanced solubility and increased absorption in the body, improving the drug’s effectiveness through co-crystallization with saccharin. Drugs belonging to the propene series also experienced improved dissolution rates through co-crystallization with nicotinamide. Additionally, a drug candidate from Boehringer Ingelheim showcased superior stability characteristics through co-crystallization. Thus, cocrystals in pharmaceutical formulations are predicted to effectively enhance the solubility, bioavailability, and stability of APIs [7,8,9].

There have been recent reports on cocrystal screening using differential scanning calorimetry (DSC). This approach involves heating binary physical mixtures of the drug and coformer using a DSC device. Detailed interpretation of the obtained DSC scans can serve as a quick screening method for cocrystal detection. This is based on the assumption that the melting point of the cocrystal differs from that of the individual components. In over 50% of cases, cocrystals have demonstrated a melting point lower than both the drug and coformer [10].

Solid-state NMR (SSNMR) is an analytical technique employed to ascertain the crystal structure of materials. This method acts as a bridge between powder X-ray diffraction (PXRD) and single crystal X-ray diffraction (SXD) analytical approaches, offering insights into the conformation of the solid and the intermolecular interactions within the solid material through the chemical shifts observed in the NMR spectrum [11]. This analytical method is valuable for predicting the crystal structure of cocrystals in powder form, where obtaining a single crystal for SXD analysis is not possible. Therefore, recent research has been conducted on the interpretation of crystal structures based on intermolecular interactions in cocrystals using solid-state NMR [12,13].

Ilaprazole (Ila) (Figure 1a) is a proton pump inhibitor (PPI) used to treat gastroesophageal reflux disease and peptic ulcers, available in the market under the trade name Noltec(Il-Yang Pharm. Co., Ltd. Yongin-si, Republic of Korea) with a 10 mg dosage of Ila [14,15,16,17,18]. These PPI typically belong to BCS class II, presenting absorption challenges due to low aqueous solubility [19,20,21,22,23]. Additionally, concerns such as the emergence of specific impurities causing color changes during a 30-day storage at room temperature or in solvents, resulting in a content decrease to as low as 84.2%, have been noted [17,19,24,25]. Despite attempts to address these stability issues through methods like salt forms [16], solvate [15], crystal form [14], and formulation change [18], satisfactory results have not been achieved.

Nevertheless, the authors of this paper, An et al., have endeavored to address this issue. They developed a 1:1 ratio Ila/Xylitol (Xyl) cocrystal with improved stability, obtaining a patent and a Chemical Abstract Service Register Number (CAS No. 2642382-31-6) [19] (Figure 1).

Therefore, this paper predicts the crystal structure of the Ila/Xyl cocrystal through solid-state CP/MAS ^13^C-NMR (SSCNMR) spectra and reports the improved stability and dissolution rate results of the Ila/Xyl cocrystal.

## 2. Materials and Methods

### 2.1. Materials

Ilaprazole (Ila) was supplied by GUJU Pharm Co. Ltd. (Hwaseong-si, Republic of Korea), while Xylitol (Xyl), Meglumine (Meg), Nicotinic acid (Nic), L-Aspartic acid (Asp), and L-Glutamic acid (Glu), as well as Acetone (ACT), Methanol (MeOH), Ethanol (EtOH), and Ethyl acetate (EA), were purchased from DaeJung Chem. Co. Ltd. (Siheung-si, Republic of Korea).

### 2.2. Ila/Xyl Cocrystal Preparation Using the Slurry Technique

Among the co-crystallization techniques, the slurry method involves forming cocrystals through solvent-mediated phase transformation (SMPT) of a mixture of API and coformer [1,26]. Therefore, it is more convenient for cocrystal production compared to the grinding method and proves to be efficient for large-scale production. In this study, the researchers employed the slurry technique to enhance the ease of production and preparation of Ila/Xyl cocrystals. The experimental procedure involved adding 10 g of Ila and 4.2 g of Xyl (1 equivalent (1 eq.)) to a 300 mL reactor, followed by the addition of 100 mL of EA. The mixture was stirred at 200 rpm for 36 h. After vacuum filtration, the resulting product underwent vacuum drying at 30 °C for 16 h. As a result, a 1:1 ratio Ila/Xyl cocrystal was formed. The stability of the Ila/Xyl cocrystal formed through the slurry method was evaluated at a temperature of 25 ± 2 °C and a relative humidity (RH) of 60 ± 5%. Furthermore, the cocrystal underwent formulation and compaction at GUJU Pharm Co. Ltd. (Seoul, Republic of Korea).

### 2.3. Differential Scanning Calorimetry (DSC)

For cocrystal screening of Ila and characterization analysis of Ila/Xyl cocrystals using DSC, a TA Instruments DSC Q25 (TA Instruments, Philadelphia, PA, USA). Additionally, samples were placed in an aluminum pan, ranging from 4.5 mg to 5 mg, and sealed using a lid. The sampling process utilized Tzero pan and lid (TA Instruments, Philadelphia, PA, USA). The DCS instrument was calibrated using indium as standards with regard to temperature and enthalpy. The thermal profile was analyzed in a nitrogen atmosphere (nitrogen flow: 50 mL/min) from 30 °C to 350 °C at a scan rate of 10 °C/min.

### 2.4. Thermogravimetric Analysis (TGA)

To perform thermogravimetric analysis of Ila/Xyl cocrystals, a TGA Q50 (TA Instruments, Philadelphia, PA, USA) was used in a nitrogen atmosphere. The temperature range was set from 30 °C to 400 °C, and the scan rate was 10 °C/min.

### 2.5. Powder X-ray Diffraction (PXRD)

Ila/Xyl cocrystal forms were analyzed with a powder X-ray diffractometer (Bruker, D8 Advance, Billerica, MA, USA) equipped with Cu Ka radiation set at 45 kV and 40 mA. The divergence and scattering slits were set as 1°, and the receiving slit was 0.2 mm. The 2θ scanning range was from 5° to 35° with a scanning rate of 3°/min (0.4 s/0.02°).

### 2.6. Solution-State Nuclear Magnetic Resonance Spectroscopy (Solution-State NMR)

For molecular structure determination, confirmation of the Ila and Xyl ratio, and atomic numbering of Ila, Xyl, and Ila/Xyl cocrystals, 1D (^1^H, ^13^C) and 2D (COSY, HSQC, HMBC) solution-state NMR spectra were obtained using a BRUKER AVANCE-800 (Billerica, MA, USA). Ila, Xyl, and Ila/Xyl cocrystals were dissolved in DMSO-d^6^ for NMR analysis.

### 2.7. Solid-State Nuclear Magnetic Resonance Spectroscopy (Solid-State CP/MAS ^13^C-NMR(SSCNMR))

The solid-state CP/MAS ^13^C-NMR spectra of Ila, Xyl and Ila/Xyl cocrystal were recorded with a 500 MHz solid-state NMR (Avance II, Bruker, Billerica, MA, USA). The spectral acquisition was achieved using the cross polarization (CP)/magic angle spinning (MAS) pulse sequence. The experimental conditions were as follows: spinning 5 KHz; pulse delay, 10 s; contact time, 2 min; and analysis time, 24 h.

### 2.8. Stability Test at 25 ± 2 °C and Relative Humidity (RH) 60 ± 5%

To assess the stability of Ila/Xyl cocrystal and Ila during storage and preservation, a stability chamber (T&H chamber, Jeio Tech, Daejeon, Republic of Korea) was utilized. The samples were stored under conditions of 25 ± 2 °C and relative humidity (RH) of 60 ± 5% for a duration of 3 months. The changes in purity were observed using high-performance liquid chromatography (HPLC) with a Waters 2487 instrument (Milford, MA, USA). Additionally, the formation of a specific impurity, ilaprazole sulfur ether (impurity B) (Figure 2), was monitored [17]. The analysis was performed using a C18 column (4.6 × 150 mm, 5 µm, Kromasil^®^, Bohus, Sweden). Mobile phase A consisted of 0.01 mol/L ammonium hydrogen phosphate (pH 6.8)/acetonitrile/methanol 90:5:5 (*v*/*v*/*v*), while mobile phase B comprised 0.01 mol/L ammonium hydrogen phosphate (pH 6.8)/acetonitrile/methanol 40:30:30 (*v*/*v*/*v*). The flow rate was set at 1.5 mL/min, wavelength at 237 nm, and run time at 50 min. Ila and impurity standards were obtained from GUJU Pharm Co. Ltd. (Seoul, Republic of Korea). A 10 mg standard solution was prepared by dissolving 1.32 g of ammonium hydrogen phosphate in 750 mL of purified water, adjusting the pH to 11.5 with a 1 mol sodium hydroxide solution. The samples were prepared by injecting 40 mL of the solution, and the retention times of Ila and impurity were confirmed. The results showed that Ila appeared at 15.7 min, and ilaprazole sulfur ether (impurity B) at 21.1 min (Appendix A). The stability chamber samples of Ila/Xyl cocrystal and Ila were prepared using the same method as the standard samples (stored at 5 °C during measurements, as Ila is unstable when dissolved in the solvent, and stability was confirmed for 24 h under 5 °C storage conditions [17].)

### 2.9. Variable-Temperature Powder X-ray Diffraction (VT-PXRD)

To confirm the melting temperature (Tm) of the Ila/Xyl cocrystal in the DSC thermal profile, a powder X-ray diffractometer (Bruker, D8 Advance, Billerica, MA, USA) with an added hot stage accessory was employed. VT-PXRD analysis was conducted at temperatures of 25 °C, 55 °C, 95 °C, and 105 °C, using the same conditions as in the PXRD analysis outlined in Section 2.5. This allowed for the verification of the Tm of the Ila/Xyl cocrystal.

### 2.10. Comparison of Dissolution Rates between Formulated Ila/Xyl Cocrystal and Ila

To compare the dissolution rates of Ila/Xyl cocrystal and Ila, formulations were obtained from GUJU Pharm Co. Ltd. (Republic of Korea), including the formulated product Noltec for Ila (Ila dosage: 10 mg) and the Ila/Xyl cocrystal formulation (Ila dosage: 10 mg) (Figure 3).

The compaction method followed the coating process according to the purification method outlined in the General Principles of Formulation of the Korean Pharmacopoeia. For each tablet, purified water (30 mg) and ethanol (30 mg) were used. This method was manufactured by GUJU Pharm Co. Ltd. (Seoul, Republic of Korea) in accordance with the standards and test methods specified by the Korean Food and Drug Administration (KDMF) for Ila (product name: Noltec). Subsequently, dissolution testing was conducted using the Dissolution Apparatus (Agilent 708-DS, Santa Clara, CA, USA) with the USP dissolution testing apparatus II Paddle (37 °C ± 0.5 °C). Dissolution tests were performed using a dissolution test solution consisting of a pH 10 buffer solution (4 L) mixed with 0.5% Polysorbate 80. Samples were collected at 0, 5, 10, 15, 30, 45, and 60 min intervals, and high-performance liquid chromatography (HPLC) (Waters 2487, Milford, MA, USA) was employed for analysis. The dissolution analysis conditions included the use of a C18 column (4.5 × 75 mm, 5 µm, Kromasil^®^, Bohus, Sweden), with the mobile phase prepared by mixing dissolution test buffer solution (3.4 g potassium dihydrogen phosphate and 0.9 g sodium hydroxide dissolved in 600 mL purified water) and acetonitrile in a 65:35 (*v*/*v*) ratio. The pH was adjusted to 8.3 using a 1 mol sodium hydroxide solution. The flow rate was set at 1.5 mL/min, wavelength at 237 nm, and run time at 50 min. To determine the peak area of Ila, a 10 mg Ila standard was dissolved in 100 mL of dissolution test buffer. This standard solution was then diluted 100 times with the dissolution test buffer, and the peak area of Ila was confirmed using this diluted standard solution. The dissolution rate was calculated using the formula: (Peak area of Ila from the sample/Peak area of Ila from the diluted standard solution) × (Amount of Ila standard collected (10 mg)/10) × (900/1000) × purity of the standard (%) (The dissolution analysis method and calculation followed the standards and testing methods of the Ila formulation product name Noltec (Ila dose: 10 mg) as specified by the Korean Food and Drug Administration (KFDA)).

## 3. Result and Discussion

### 3.1. Cocrystal Screening of Ila Using DSC

The possibility of screening the formation of cocrystals through the heating of API (Active Pharmaceutical Ingredient) and coformer physical mixtures using DSC (Differential Scanning Calorimetry) has been reported. This is attributed to the different melting points of the API, coformer, and cocrystal [10]. Therefore, in this study, to address issues related to the poor stability and release rate during the storage and handling processes of Ila, we aimed to develop cocrystals of Ila with enhanced stability. For this purpose, five potential coformers rich in functional groups such as OH and O, capable of forming hydrogen bonds with Ila (pKa 10.10 [27]), were selected, as presented in Table 1 and Appendix A [13].

After preparing physical mixtures of the coformers and Ila in a 1:1 ratio, as indicated in Table 1 and Appendix A, DSC screening was conducted, and the results are presented in Figure 4. In Figure 4a,b, the DSC thermal profiles of the Ila/Asp and Ila/Glu physical mixtures did not show any new exothermic or endothermic peaks; instead, the profiles exhibited a mixed form. Therefore, Asp and Glu were predicted to have no potential for cocrystal formation with Ila and were excluded.

In Figure 4c, the DSC thermal profile of the Ila/Meg physical mixture showed an endothermic peak around 128 °C and a new exothermic peak around 150 °C. The endothermic peak at 130 °C appeared slightly ahead of the endothermic peak of Meg at 130 °C, and the endothermic peak corresponding to Ila at 162 °C was absent. In Figure 4d, the DSC thermal profile of the Ila/Nic physical mixture did not exhibit predicted endothermic and exothermic peaks associated with Ila and Nic. Instead, a new endothermic peak appeared at approximately 138 °C, and a new exothermic peak emerged at around 147 °C. The presence of new endothermic and exothermic peaks in the DSC thermal profiles of physical mixtures indicates the potential for cocrystal formation. This is because the formation of exothermic peaks in the DSC thermal profiles can predict phase transformation into a cocrystal, and changes in endothermic peaks can be indicative of variations in the melting point [10].

Lastly, in Figure 4e, the DSC thermal profile of the Ila/Xyl physical mixture exhibited a matched endothermic peak of Xyl at 98 °C. However, new small endothermic and exothermic peaks appeared at 147 °C and 153 °C, respectively. Furthermore, the endothermic peak corresponding to Ila at 162 °C was not observed. These results suggest the potential for new interactions or changes in the melting point, providing insight into the possibility of cocrystal formation [10].

Thus, with the predicted potential for cocrystal formation through DSC screening, co-crystallization attempts were made using Meg, Nic, and Xyl via the co-crystallization technique known as slurry.

### 3.2. Co-Crystallization of Ila Using the Slurry Technique

Based on the DSC thermal profile results in Figure 4, coformers Meg, Nic, and Xyl, which showed potential for cocrystal formation with Ila, were selected. Ila (1 g) and the chosen coformer (1 eq.) were individually added to MeOH, ACT, EtOH, and EA solvents (10 mL each) and stirred at room temperature for 24 h to assess the potential for co-crystallization. In MeOH, ACT, and EtOH solvents, either Ila or the coformer dissolved, making it impossible to obtain a solid or obtaining only a small amount. Therefore, these solvents were excluded. However, in EA, a significant amount of solid was obtained. EA was chosen as the solvent, and Ila (10 g) was used, stirring at room temperature for 36 h, followed by PXRD analysis.

The results indicated that both Ila/Meg and Ila/Nic exhibited PXRD patterns similar to the physical mixture, suggesting no potential for co-crystal formation (Appendix A). However, for Ila/Xyl, the PXRD pattern differed significantly from the PXRD patterns of Ila, Xyl, and the physical mixture.

Figure 5 shows the PXRD patterns of Ila, Xyl, Ila/Xyl physical mixture, and the Ila/Xyl obtained through co-crystallization. In the PXRD pattern of the co-crystallized Ila/Xyl, peaks at 2θ were observed at 6.83°, 12.57°, 15.78°, 18.06°, 19.42°, 19.74°, and 22.46°. In contrast, the 2θ values for the Ila/Xyl physical mixture were 7.34°, 13.02°, 16.15°, 18.55°, 22.84°, 25.82°, and 31.81°. Additionally, Xyl exhibited peaks at 2θ values of 14.85°, 17.91°, 22.84°, and 24.96°, while Ila showed peaks at 2θ values of 7.29°, 13.02°, 16.20°, 18.50°, 21.50°, and 25.84°. Therefore, it is predicted that the Ila/Xyl crystal obtained through co-crystallization has a new crystal structure, indicating the potential for cocrystal formation. However, further analysis is needed for a conclusive result.

### 3.3. Prediction of Intermolecular Interaction in Ila/Xyl Cocrystal Using Solid-State CP/MAS ^13^C-NMR (SSCNMR)

To predict that the Ila/Xyl crystal obtained through co-crystallization is indeed a cocrystal, attempts were made to analyze it using SSCNMR. To interpret the analyzed SSCNMR spectra, solution-state NMR 1D (1H, 13C) and 2D (1H-1H COSY, 1H-13C HSQC, and 1H-13C HMBC) were analyzed. The molecular structures of Ila, Xyl, and Ila/Xyl were interpreted, and the positions of carbon (C) and hydrogen (H) were confirmed by numbering them according to the molecular structures of Ila and Xyl, as seen in Figure 1. This information served as the basis for interpreting the SSCNMR spectrum analysis (Appendix A).

Figure 6 presents the SSCNMR spectra of Ila, Xyl, Ila/Xyl physical mixture, and the Ila/Xyl crystal obtained through co-crystallization. As observed in the SSCNMR spectra of Figure 6, it can be predicted that the spectrum of the Ila/Xyl crystal obtained through co-crystallization shows differences in chemical shift and split of peaks compared to the spectra of Ila, Xyl, and the Ila/Xyl physical mixture.

In Figure 6a, when comparing the spectrum of the Ila/Xyl crystal obtained through co-crystallization (shown in blue) with the spectra of Ila, Xyl, and the Ila/Xyl (1:1) physical mixture, a noticeable shift in C15 to a more upfield position at 134.83 ppm can be observed from the Ila molecular structure in Figure 1a. This shift is predicted to result from the benzimidazole N-H(18) in the Ila structure acting as a hydrogen bond donor. The enhanced electron density due to acting as a hydrogen bond donor would affect the resonance of C15 at the para position, causing it to shift upfield.

Furthermore, in Figure 6a of the Ila/Xyl crystal obtained through co-crystallization, changes in the split of C14 are observed at 154.13 ppm and 151.2 ppm. This change is anticipated to be influenced by the benzimidazole N-H(18) in the Ila structure acting as a hydrogen bond donor and the sulfoxide S=O acting as a hydrogen bond acceptor, as indicated in Figure 1a. Evidence supporting the sulfoxide S=O as a hydrogen bond acceptor can be inferred from the changes in the chemical shift of C3 in Figure 6b.

Additionally, significant changes in the chemical shifts of C2 and C12 in Figure 6a are predicted to be due to the nitrogen (N) of the pyridine in the Ila structure acting as a hydrogen bond acceptor, leading to changes in the chemical shift of C2 and C12 through resonance effects.

In Figure 6b, the SSCNMR spectrum shows Xyl C1, C2, and C3 appearing in the range of 60 ppm to 75 ppm, as indicated in Figure 1b. However, it is observed in this spectrum that there is no significant difference in chemical shift and split. This lack of difference is attributed to molecular structures with hydroxy groups (OH), such as Xyl, being capable of acting as both hydrogen bond acceptors and donors simultaneously. Therefore, such molecular structures can exhibit both intramolecular and intermolecular interactions, resulting in a consistent electron distribution. As a consequence, the SSCNMR spectrum in Figure 6b is predicted to show no significant differences in chemical shift and split due to these interactions [29].

The results of the SSCNMR spectrum in Figure 6 predicts the presence of hydrogen bonds, specifically between the N-H (donor) of Ila and the O (acceptor) of Xyl, the S=O (acceptor) of Ila and the O-H (donor) of Xyl, as well as the N (acceptor) of Ila and the O-H (donor) of Xyl.

SSCNMR is an effective tool for predicting intermolecular interactions and conformational changes through changes in chemical shift and splitting. It becomes particularly valuable when single crystals cannot be precipitated, and SXD analysis is not feasible for predicting crystal structures [11,12,13]. Therefore, based on the SSCNMR results in Figure 6, it is predicted that the Ila/Xyl crystal obtained through co-crystallization can be identified as a cocrystal. The reason Ila and Xyl could form a cocrystal in a slurry, i.e., a solvent, is attributed to the small acidity difference between Ila and Xyl as indicated in Table 1. The low acidity prevents ionization between the two molecules in the solvent environment, and the abundance of functional groups capable of forming intermolecular interactions further supports this prediction.

Furthermore, the formation of Ila/Xyl cocrystal in a 1:1 ratio was predicted through solution-state ^1^H-NMR spectra (Figure 7). This result was further refined and confirmed through ^1^H-NMR analysis (Appendix A) and 2D ^1^H-^1^H COSY analysis (Appendix A).

### 3.4. Thermal Analysis of Ila/Xyl Cocrystal

Figure 8 shows the DSC (Differential Scanning Calorimetry) heat curves and TGA (Thermogravimetric Analysis) results for Ila/Xyl cocrystal, Ila, and Xyl. In the results, the Ila/Xyl cocrystal exhibits an endothermic onset temperature at 92.11 °C, while Ila shows an endothermic onset temperature at 158.35 °C, and Xyl at 93.21 °C. Additionally, in TGA, the Ila/Xyl cocrystal starts to experience mass loss due to thermal decomposition around 140 °C, Ila at 155 °C, and Xyl at approximately 220 °C.

Despite the fact that there is not a significant difference in the endothermic temperatures between Ila/Xyl cocrystal and Xyl, to precisely determine the melting temperature, VT-PXRD analysis was conducted to predict its accuracy.

Figure 9 shows the VT-PXRD (Variable Temperature Powder X-ray Diffraction) patterns of Ila/Xyl cocrystal analyzed in the temperature range from 25 °C to 105 °C. This analysis was conducted to predict that the DSC endothermic onset temperature observed at 92.11 °C in Figure 8 corresponds to the melting temperature.

In the results, at 25 °C and 55 °C, the PXRD patterns maintain the same 2θ values, indicating the stability of the cocrystal. However, at 95 °C, which corresponds to the endothermic temperature observed in Figure 8, the PXRD pattern is nearly disappearing. Furthermore, at 105 °C, no distinct peaks related to 2θ are observed, indicating the complete absence of the characteristic PXRD pattern. This suggests that from 95 °C onward, the Ila/Xyl cocrystal melts, and the PXRD pattern is not evident. Therefore, the temperature of the DSC endothermic peak in Figure 8, representing the Ila/Xyl cocrystal, can be predicted as the melting temperature by observing the VT-PXRD pattern changes.

The analysis results from Table 1 and Table 2 and Figure 5, Figure 6, Figure 7, Figure 8 and Figure 9 predict the formation of a 1:1 ratio cocrystal between Ila and Xyl, and the characterization of this cocrystal has been confirmed.

### 3.5. Stability Evaluation for Room Temperature Storage at 25 ± 2 °C and RH 60 ± 5% of Ila/Xyl Cocrystal

Proton pump inhibitors (PPIs) such as Ila often face challenges in stability, with issues like discoloration and the generation of specific impurities, such as ilaprazole sulfur ether (impurity B in Figure 2), during storage at room temperature for 30 days. This results in a decrease in purity to below 95%, posing problems for its use as an active pharmaceutical ingredient (API) and formulation development. Consequently, storing Ila at 5 °C has been reported to the Korea Ministry of Food and Drug Safety (KMFDS) [17,19,24,25].

APIs with low stability present difficulties in formulation development, product handling, dosage form issues, and analytical method challenges. Cocrystals, as crystalline solids formed by the intermolecular interaction between an API and a coformer, can share the advantages of the coformer with the API. Consequently, they offer a promising solution to overcome the current issues with Ila, providing an optimal crystal structure that enhances stability, solubility, and dissolution rates [1,2,3,4,5].

According to the standards and test methods specified by the Korean Food and Drug Administration (KFDA) for Ila, the impurity ilaprazole sulfur ether (impurity B (Figure 2)) should be present in an amount of no more than 0.15%. Unknown impurities should be below 0.1%, and the total impurity content should be less than 1.0% (with a purity of 99% or more).

Figure 10 depicts the results of stability testing conducted by the researchers to predict whether the newly developed Ila/Xyl cocrystal improves the low stability associated with Ila during room temperature storage. Ila and Ila/Xyl cocrystal were subjected to stability chambers under conditions of 25 ± 2 °C and RH 60 ± 5% for three months, and their purity was assessed. As observed in Figure 10, the Ila/Xyl cocrystal maintains its high purity, with minimal changes, from an initial purity of 99.8% to 99.8% after one month and 99.75% after three months. In contrast, Ila shows a decline in purity from an initial 99.8% to 94% after one month and further to 90% after three months, highlighting the improved stability of the cocrystal compared to Ila alone (Figure 10 was tested with six repetitions of the sample, and the relative standard deviation was less than ± 3% during these trials).

Table 2 presents the results of HPLC analysis to determine whether a specific impurity, ilaprazole sulfur ether (impurity B, Figure 2 [17]), was generated in the samples over the 3-month period. The analysis was conducted by confirming the retention time of ilaprazole sulfur ether using HPLC (Appendix A).

As shown in Table 2, the Ila/Xyl cocrystal exhibited minimal generation of the impurity ilaprazole sulfur ether (Figure 2) over the storage period. The initial content was 0.004%, increasing to 0.023% after 1 month and 0.030% after 3 months, remaining well below 0.1%. In contrast, Ila showed a significant increase in a specific impurity, ilaprazole sulfur ether, starting from an initial content of 0.032% to 0.82% after 1 month and further to 2.28% after 3 months.

Molecules with structures like Xyl can engage in both intramolecular and intermolecular interactions, acting as both hydrogen bond acceptors and donors. This property contributes to the overall stability of the molecule [29,30]. The improved stability of Ila through cocrystallization with Xyl, as observed in Figure 10 and Table 2, is attributed to the intermolecular interaction with Xyl as a coformer, leading to the formation of the cocrystal.

Therefore, it is predicted that the Ila/Xyl cocrystal, with its enhanced stability, can be stored and maintained at room temperature. The cocrystal demonstrates the potential for Ila to serve as an improved solid-state form with enhanced stability as an active pharmaceutical ingredient (API).

### 3.6. Comparison of Dissolution Rates between Ila/Xyl Cocrystal Formulation and Ila Noltec Formulation

Ila is commercially available under the product name Noltec at a dosage of 10 mg. However, due to the expiration of the patent, efforts have been made to launch Ila as a generic drug by developing salt forms [16], solvates [15], crystal forms [14], and formulation changes [18] to achieve an equivalent or higher dissolution rate. Despite these efforts, generic market entry has proven to be challenging.

It is understood that, according to the standards and testing methods outlined by the KFDA for Noltec, it is specified that Ila’s dissolution rate should reach 80% or more 30 min after administration. Consequently, experiments were conducted to verify whether the dissolution rate of Ila/Xyl cocrystal achieves 80% or more after 30 min. Figure 11 illustrates the dissolution rate results for the Ila/Xyl cocrystal formulated at a dosage of 10 mg (as shown in Figure 3) and the Ila Noltec formulation (dosage: 10 mg). In this graph, the Ila/Xyl cocrystal reaches 54.33% at 15 min, 96.49% at 30 min, and 100% at 45 min. In comparison, Ila achieves 33.67% at 15 min, 94.45% at 30 min, 98.15% at 45 min, and 100% at 60 min (the test was conducted with six repetitions, and the coefficient of variation was less than ±5%). Although the dissolution rate of the Ila/Xyl cocrystal may be slightly faster than that of Ila, considering the margin of error, it is predicted to be roughly equivalent. Despite not achieving an equivalent or higher dissolution rate, the Ila Noltec formulation has faced challenges in the generic drug market. However, the Ila/Xyl cocrystal, with the advantage of stable storage at room temperature, is expected to expedite the generic drug launch process.

## 4. Conclusions

In this study, we aimed to address the stability issues of Ila, a medication used for the treatment of gastroesophageal reflux disease (GERD) and peptic ulcers, marketed under the brand name Noltec (10 mg). The challenge was to improve its stability at room temperature and during storage. As a result, we successfully formed a cocrystal of Ila and Xyl in a 1:1 ratio, which can be stored at room temperature without the need for refrigeration. The predicted crystal structure of the Ila/Xyl cocrystal was confirmed through the analysis of solid-state cross-polarization/magic angle spinning carbon-13 nuclear magnetic resonance (SSCNMR) spectra, providing insights into the intermolecular interactions between Ila and Xyl.

The enhanced stability of the Ila/Xyl cocrystal was demonstrated through a three-month stability evaluation under conditions of 25 ± 2 °C and relative humidity (RH) 65 ± 5%. The cocrystal maintained a purity of 99.8% throughout the storage period, whereas the purity of Ila decreased to 90% after three months, accompanied by the significant generation of a specific impurity, ilaprazole sulfur ether. The improved stability of the Ila/Xyl cocrystal was attributed to the inherent stability of the coformer Xyl, suggesting that the intermolecular interactions between Ila and Xyl played a crucial role.

Furthermore, dissolution rate tests comparing the Ila/Xyl cocrystal formulation (10 mg) and the commercial Ila formulation Noltec (10 mg) were conducted at pH 10. The results indicated that the Ila/Xyl cocrystal exhibited a dissolution rate comparable to or slightly faster than that of Ila Noltec. This suggests that the cocrystal formulation, with its enhanced stability and potential for room temperature storage, could expedite the generic drug release process for Ila, addressing the challenges faced by the generic pharmaceutical industry in developing equivalent dissolution rates.

In conclusion, the development of the Ila/Xyl cocrystal not only overcomes the stability issues associated with Ila, allowing for room temperature storage, but also offers advantages in formulation development, product handling, dosing, and analytical testing. The cocrystal, with its improved stability and dissolution characteristics, represents an optimal crystalline structure, providing a promising solution for the challenges associated with Ila as an active pharmaceutical ingredient (API). Therefore, the development of cocrystals is anticipated to play a crucial role in future pharmaceutical development, not only for enhancing absorption rates in drugs with solubility issues but also for improving the physicochemical properties of APIs through the unique characteristics of crystal structures.

## Figures and Tables

**Figure 1 pharmaceutics-16-00122-f001:**
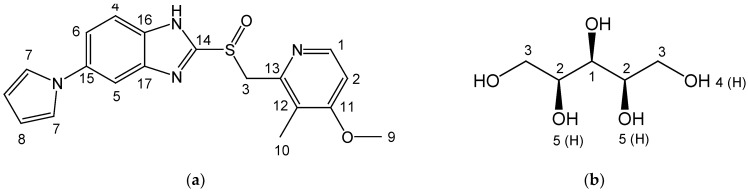
Molecular structure with atom numbers: (**a**) Ilaprazole (Ila); (**b**) Xylitol (Xyl).

**Figure 2 pharmaceutics-16-00122-f002:**
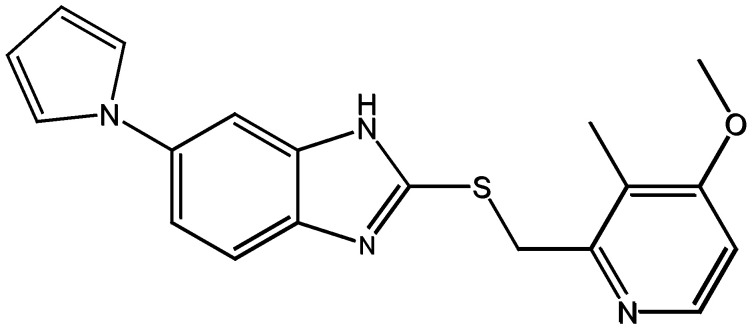
Molecular structure of ilaprazole sulfur ether (impurity B) [17].

**Figure 3 pharmaceutics-16-00122-f003:**
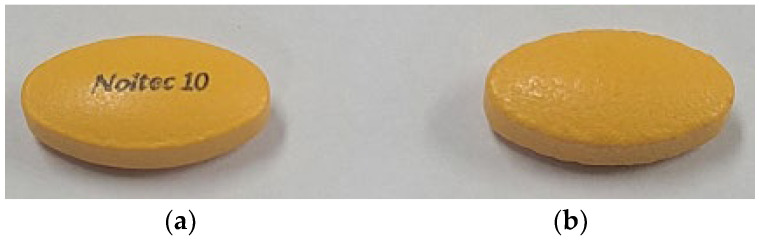
Formulation images of Ila and Ila-Xyl cocrystal: (**a**) Ila formulation (brand name: Noltec (10 mg as Ila)); (**b**) Ila/Xyl cocrystal (10 mg as Ila).

**Figure 4 pharmaceutics-16-00122-f004:**
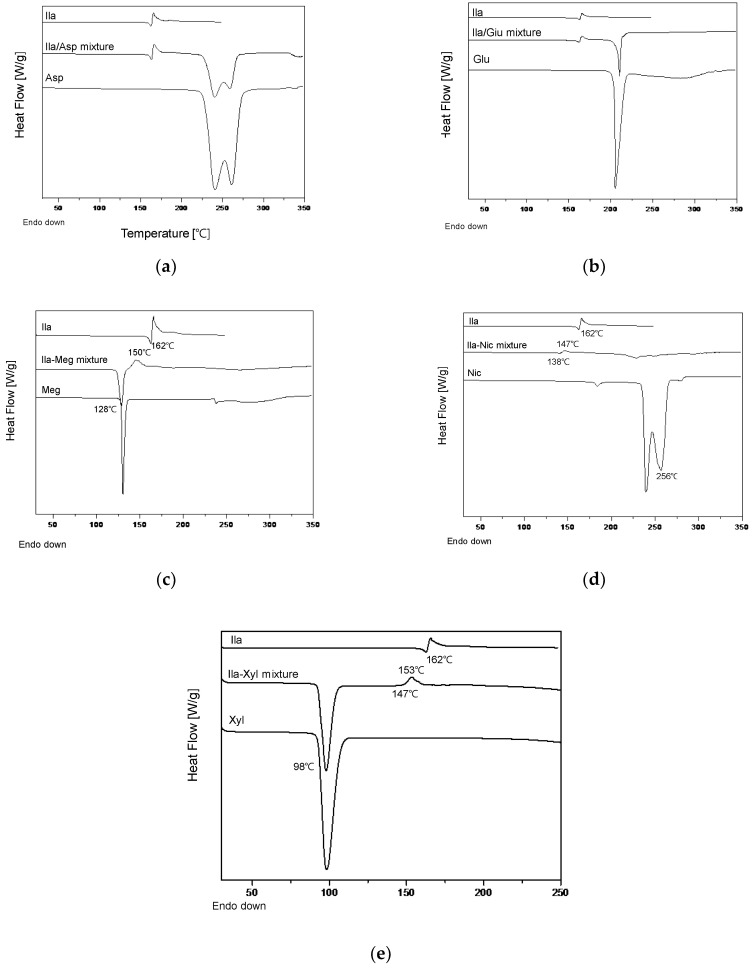
Ila cocrystal screening using DSC: (**a**) Ila, Asp and Ila/Asp (1:1) physical mixture; (**b**) Ila, Glu and Ila/Glu (1:1) physical mixture; (**c**) Ila, Meg and Ila/Glu (1:1) physical mixture; (**d**) Ila, Nic and Ila/Nic (1:1) physical mixture; and (**e**) Ila, Xyl and Ila/Xyl (1:1) physical mixture.

**Figure 5 pharmaceutics-16-00122-f005:**
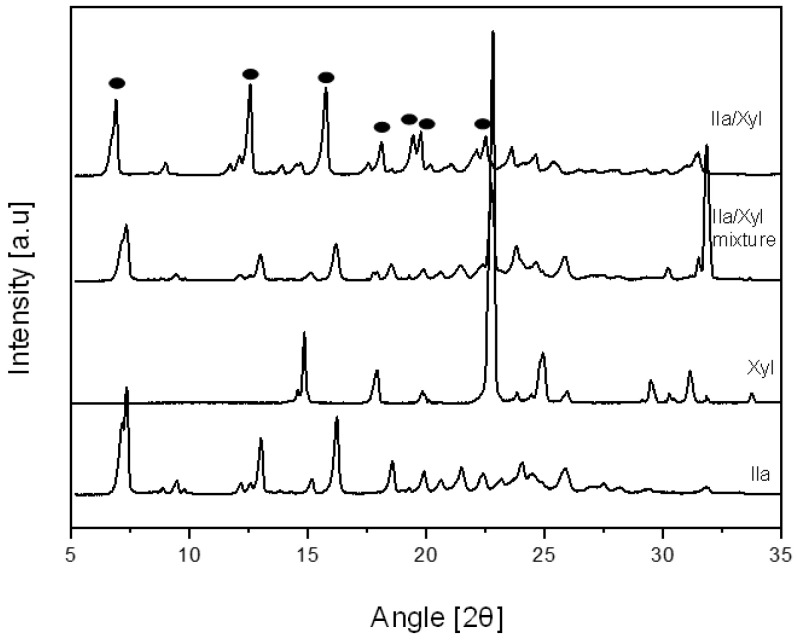
PXRD pattern of Ila, Xyl, Ila/Xyl physical mixture and Ila/Xyl crystal obtained by co-crystallization.

**Figure 6 pharmaceutics-16-00122-f006:**
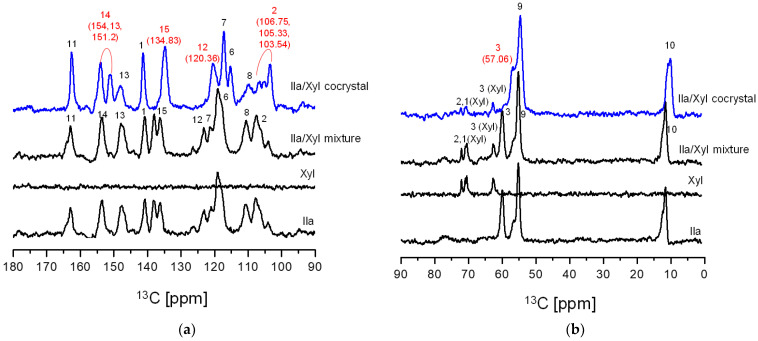
Ila/Xyl (1:1) cocrystal, Ila, Xyl, and Ila/Xyl (1:1) physical mixture’s solid-state CP/MAS 13C-NMR: (**a**) 90–180 ppm; (**b**) 0 ppm–90 ppm (numbers on the peaks are related to numbers in Figure 1 structure location). (The spectrum of the Ila/Xyl cocrystal is displayed in blue).

**Figure 7 pharmaceutics-16-00122-f007:**
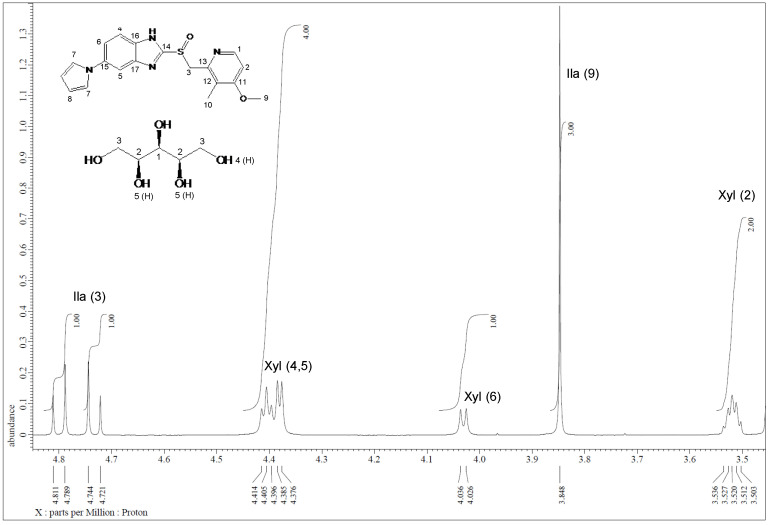
Integration of solution-state ^1^H-NMR spectrum of Ila/Xyl cocrystal (DMSO-d^6^).

**Figure 8 pharmaceutics-16-00122-f008:**
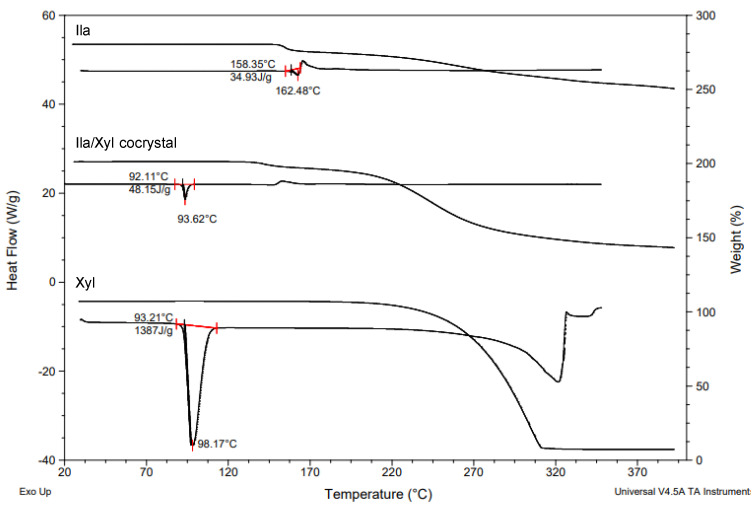
TGA-DSC curves of Ila/Xyl cocrystal, Ila and Xyl (heating rate 10 °C/min) (Indicate the integrated value of the endothermic curve in red on the DSC heat profile).

**Figure 9 pharmaceutics-16-00122-f009:**
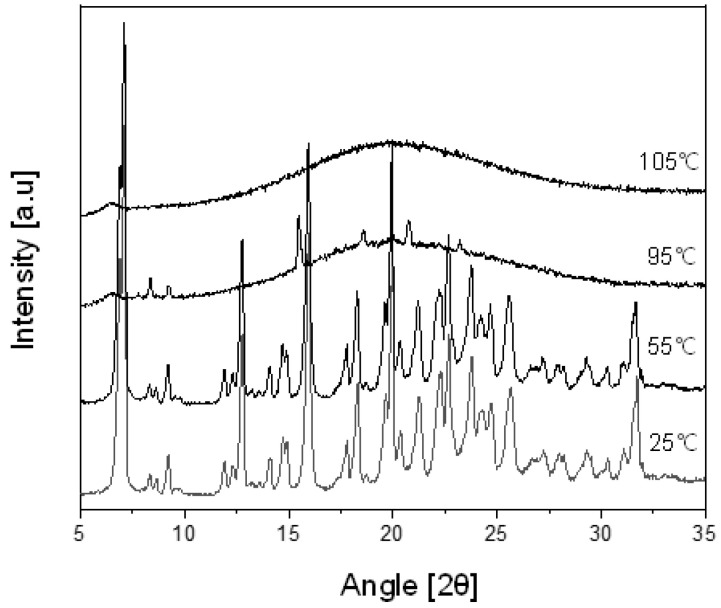
VT-PXRD pattern of Ila/Xyl cocrystal according to temperature change.

**Figure 10 pharmaceutics-16-00122-f010:**
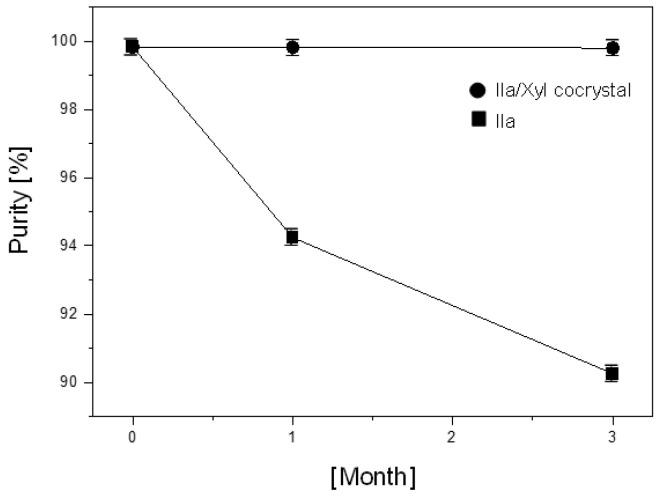
Purity results analyzed after 3 months of storage at 25 ± 2 °C and RH 60 ± 5% for Ila and Ila/Xyl cocrystal (error bar: test for six repetitions).

**Figure 11 pharmaceutics-16-00122-f011:**
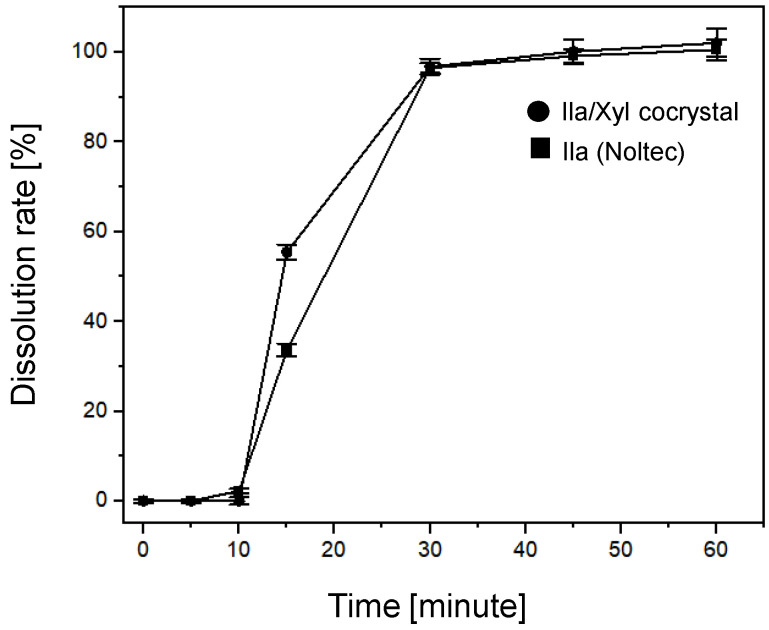
Dissolution rate test at pH 10 for Ila/Xyl cocrystal formulation (Dosage: 10 mg) and Ila Noltec formulation (dosage: 10 mg) (refer to the formulation images in Figure 3) (error bar: test for six repetitions).

**Table 1 pharmaceutics-16-00122-t001:** Hydroxy group used as a coformer and their pKa.

Coformer (Equivalent)	pKa (Ref. [28])
L-Aspartic acid (Asp) (1 eq.)	1.88 3.65
L-Glutamic acid (Glu) (1 eq.)	2.19 4.25
Meglumine (Meg) (1 eq.)	9.52
Nicotinic acid (Nic) (1 eq.)	4.85
Xylitol (Xyl) (1 eq.)	12.76

**Table 2 pharmaceutics-16-00122-t002:** Results of ilaprazole sulfur ether impurity analysis after 3 months of storage at 25 ± 2 °C and RH 65 ± 5% for Ila and Ila/Xyl cocrystal.

Month	Ila	Ila/Xyl Cocrystal
0	0.032%	0.004%
1	0.82%	0.023%
3	2.28%	0.030%

## Data Availability

The data presented in this study are available in this article (and Appendix A).

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
