# Peer review of "Investigation of the Storage and Stability as Well as the Dissolution Rate of Novel Ilaprazole/Xylitol Cocrystal"

_pharmaceutics, 2024, doi:10.3390/pharmaceutics16010122_

Round 1

Reviewer 1 Report

Comments and Suggestions for Authors

The manuscript entitled “Investigation of the storage and stability as well as the dissolution rate of novel Ilaprazole/Xylitol Cocrystal” presents an interesting cocrystal screening results. The authors thoroughly analyzed important from the new drug form development perspective parameters, such as stability and dissolution rate. In my opinion the study is valuable and innovative, however I have several minor comments listed below:

(I) 2.3. Differential Scanning Calorimetry (DSC): I suggest adding information about the calibration of the DSC device (types of standards used), gas flow and the type of crucibles used.

(II) Figure 4: The “endo up” or “endo down” information should be included in the figure (heat flow axis) or figure caption.

(III) 3.4. Thermal analysis of Ila/Xyl cocrystal: The authors provided melting point values determined by the DSC method. It is unclear whether these values represent the onset or peak values. To ensure clarity, please specify whether the provided values are onset values. Of course onset values are more appropriate.

(IV) Figure 10, Figure 11: The number of replicates of the experiments is not provided in the figure caption. I recommend presenting the error bars in the figures (standard deviations).

Author Response

Reviewer 1

We thank the Reviewer 1 for the valuable comments about our manuscript. We tried to revise the manuscript as much as possible in line with your comments.

2.3. Differential Scanning Calorimetry (DSC): I suggest adding information about the calibration of the DSC device (types of standards used), gas flow and the type of crucibles used.

- Thank you for your evaluation. Sincerely appreciating your feedback, we have revised the manuscript according to the comments from Reviewer 1, and highlighted the changes in red. The modified content is as follows:

For cocrystal screening of Ila and characterization analysis of Ila/Xyl cocrystals using DSC, a TA Instruments DSC Q25 (TA Instruments, Philadelphia, PA, USA). Additionally, samples were placed in an aluminum pan, ranging from 4.5 mg to 5 mg, and sealed using a lid. The sampling process utilized Tzero pan and lid (TA Instruments, Philadelphia, PA, USA). DCS instrument was calibrated using indium as standards with regard to temperature and enthalpy. The thermal profile was analyzed in a nitrogen atmosphere (nitrogen flow: 50mL/min) from 30 ℃ to 350 ℃ at a scan rate of 10 ℃/min.

Figure 4: The “endo up” or “endo down” information should be included in the figure (heat flow axis) or figure caption.

 -  I have revised Figure 4 in accordance with the feedback provided. Kindly review the manuscript.

3.4. Thermal analysis of Ila/Xyl cocrystal: The authors provided melting point values determined by the DSC method. It is unclear whether these values represent the onset or peak values. To ensure clarity, please specify whether the provided values are onset values. Of course onset values are more appropriate.

 - Thank you for your valuable feedback. In line with these comments, I have revised the content of the manuscript to "endothermic onset temperature.

Figure 10, Figure 11: The number of replicates of the experiments is not provided in the figure caption. I recommend presenting the error bars in the figures (standard deviations).

 - I have thoroughly revised both Figure 10 and 11, incorporating the constructive feedback from Reviewer 1. Thank you once again for your valuable insights.

Sincerely yours,

Ji-Hun An, Ph.D.

CEO

Unicel Lab

Uiwang, Gyeonggi, 16079, Republic of Korea

Reviewer 2 Report

Comments and Suggestions for Authors

1.       What is the main question addressed by the research?

Manuscript entitled “Investigation of the storage and stability as well as the dissolution rate of novel Ilaprazole/Xylitol Cocrystal” is very interesting and presents detailed research.

2.       Do you consider the topic original or relevant in the field? Does it
address a specific gap in the field?

3.       The article is very interesting, current and original.

The manuscript deals with a narrow modern topic that may be of interest to many scientists.

4.  What does it add to the subject area compared with other published
material?

The manuscript deals with a narrow modern topic that may be of interest to many scientists.

5. What specific improvements should the authors consider regarding the
methodology? What further controls should be considered?

Did the authors make only one formula? Why? In my opinion Authors should be assessed drug loading in the prepared co-crystals.

6. Are the conclusions consistent with the evidence and arguments presented
and do they address the main question posed?

Conclusions are well prepared, but in my opinion in to long.

7. Are the references appropriate?

Yes, references are appropriate and actual

8. Please include any additional comments on the tables and figures

Tables and Figures are well prepared and legible. Figures are clear and interesting, good quality

Author Response

Reviewer 2

We thank the Reviewer 2 for the valuable comments about our manuscript. I sincerely appreciate your interest in our research results and thank you once again for providing positive feedback. I would like to express my gratitude to Reviewer 2 for taking the time to review our manuscript.

Reviewer 3 Report

Comments and Suggestions for Authors

The supplied manuscript “Investigation of the storage and stability as well as the dissolution rate of novel Ilaprazole/Xylitol Cocrystal” examines the stability and dissolution rate of Ilaprazole/Xylitol cocrystal. It explores the use of Xylitol as a coformer to form cocrystals with Ilaprazole, a medication used for treating gastric ulcers and reflux esophagitis. The study finds that the Ilaprazole/Xylitol cocrystal offers improved stability and dissolution rate, eliminating the need for refrigeration and enhancing the drug's effectiveness. This advancement is significant for pharmaceutical formulation development, addressing challenges related to Ilaprazole's stability at room temperature.

However the authors should address some minor observations prior to acceptance for publication which are detailed below.

The authors mention the importance of functional groups present in the coformer yet no evidence is provided and the figure S3 does not label each of the coformers and needs to be read in conjunction with Table 1 in the main paper. Labelling the coformers would be advantageous and recommended.

Section 2.2 - The authors have not stated the recovery/ extraction / drying technique used in the formulation and what material was used for the physical characterisation was it the cocrystal direct from the slurry or the formulated tablet shown in Fig 3.  Also details of the formulation are required to show that they are equivalent to the commercial product Section 2.10

Line 400 – The authors in the materials and methods section have stated the storage condition is  25°C / 60  % rh  which is the typical ICH long term condition and  from Line 400  the authors refer to the  25°C / 65  % rh  . Can the authors confirm the actual conditions and is 3 months sufficient for a stability study.

The authors have not given the anticipated specification limits or acceptance criteria for the CQAs investigated (potency/impurity and dissolution) which is important to determine whether stability will be maintained.

It would also be interesting and beneficial to see the stability of the proposed formulation in comparison to those stored under refrigerated conditions.

There is an absence of any statistical evaluation of the quantitative results which is necessary for publication including stating the number of replicate measurements for each sample.

Finally, I would urge the authors to reconsider the title as the stability and dissolution aspect are only small aspects of the research undertaken (while providing the headline results) in the manuscript and more focus should be directed to the physico-chemical characterisation of the cocrystal.

Comments on the Quality of English Language

In summary, the manuscript is comprehensive and detailed in its content, only minor editing of the English style is required. Some specific examples are listed below.

Line 60 - Some API face challenges such as low solubility, resulting in decreased bioavailability and stability issues. This sentence is not grammatically correct and  as "API" is plural here, it should be "Some APIs face challenges..."

Line  69 In this manner, cocrystal in pharmaceutical formulations is predicted to be an effective method for enhancing solubility, bioavailability, and stability of API [7-9].

Consider rephrasing for clarity, such as "Thus, cocrystals in pharmaceutical formulations are predicted to effectively enhance the solubility, bioavailability, and stability of APIs."

Noltec is spelled incorrectly (Noitec) in Figure 11

Author Response

Reviewer 3

We thank the Reviewer 3 for the valuable comments about our manuscript. We tried to revise the manuscript as much as possible in line with your comments.

The authors mention the importance of functional groups present in the coformer yet no evidence is provided and the figure S3 does not label each of the coformers and needs to be read in conjunction with Table 1 in the main paper. Labelling the coformers would be advantageous and recommended.

 - Thank you sincerely for pointing out the shortcomings in our manuscript. Consequently, we have actively incorporated the feedback from Reviewer 3 and made revisions to Figure S3. The modified sections are highlighted in red, and the details are as follows:

Figure S3. The molecular structures of the coformers presented in Table 1; (a) L-Aspartic acid (Asp); (b) L-Glutamic acid (Glu); (c) Meglumine (Meg); (d) Nicotinic acid (Nic); (e) Xylitol (Xyl)

Section 2.2 - The authors have not stated the recovery/ extraction / drying technique used in the formulation and what material was used for the physical characterisation was it the cocrystal direct from the slurry or the formulated tablet shown in Fig 3.

 - Once again, thank you for your feedback. We have made the necessary revisions to the manuscript, marked in red. The modified content is as follows:

2.2. Ila/Xyl Cocrystal Preparation Using the Slurry Technique

Among the co-crystallization techniques, the slurry method involves forming cocrystals through solvent-mediated phase transformation (SMPT) of a mixture of API and coformer [1, 26]. Therefore, it is more convenient for cocrystal production compared to the grinding method and proves to be efficient for large-scale production. In this study, the researchers employed the slurry technique to enhance the ease of production and preparation of Ila/Xyl cocrystals. The experimental procedure involved adding 10g of Ila and 4.2g of Xyl (1 equivalent (1eq.)) to a 300mL reactor, followed by the addition of 100mL of EA. The mixture was stirred at 200rpm for 36 hours. After vacuum filtration, the resulting product underwent vacuum drying at 30°C for 16 hours. As a result, a 1:1 ratio Ila/Xyl cocrystal was formed. The stability of the Ila/Xyl cocrystal formed through the slurry method was evaluated at a temperature of 25±2°C and a relative humidity (RH) of 60±5%. Furthermore, the cocrystal underwent formulation and compaction at GUJU Pharm Co. Ltd. (Korea), as depicted in Figure 3.

Also details of the formulation are required to show that they are equivalent to the commercial product Section 2.10

- Once again, thank you for your valuable feedback. We have made the necessary revisions to the manuscript, marked in red. The modified content is as follows:

2.10. Comparison of Dissolution Rates between Formulated Ila/Xyl Cocrystal and Ila

The compaction method followed the coating process according to the purification method outlined in the General Principles of Formulation of the Korean Pharmacopoeia. For each tablet, purified water (30mg) and ethanol (30mg) were used. This method was manufactured by GUJU Pharm Co. Ltd. (Korea) in accordance with the standards and test methods specified by the Korean Food and Drug Administration (KDMF) for Ila (product name: Noltec).

Line 400 – The authors in the materials and methods section have stated the storage condition is 25°C / 60 % rh  which is the typical ICH long term condition and  from Line 400  the authors refer to the  25°C / 65  % rh  . Can the authors confirm the actual conditions and is 3 months sufficient for a stability study.

- Thank you for your valuable feedback. In line with these comments, I have revised the content of the manuscript to "25±2°C and RH 60±5%.

The authors have not given the anticipated specification limits or acceptance criteria for the CQAs investigated (potency/impurity and dissolution) which is important to determine whether stability will be maintained.

- Once again, thank you for your valuable feedback, Reviewer 3. Consequently, we have actively incorporated your suggestions and made revisions to the manuscript. The modified sections are highlighted in red, and the details are as follows:

3.5 Stability Evaluation for Room Temperature Storage at 25±2°C and RH 60±5% of Ila/Xyl Cocrystal

According to the standards and test methods specified by the Korean Food and Drug Administration (KFDA) for Ila, the impurity ilaprazole sulfur ether (impurity B (Figure 2)) should be present in an amount of no more than 0.15%. Unknown impurities should be below 0.1%, and the total impurity content should be less than 1.0% (with a purity of 99% or more).

3.6. Comparison of Dissolution Rates between Ila/Xyl Cocrystal Formulation and Ila Noltec Formulation

I understand that, according to the standards and testing methods outlined by the KFDA for Noltec, it is specified that Ila's dissolution rate should reach 80% or more 30 minutes after administration. Consequently, experiments were conducted to verify whether the dissolution rate of Ila/Xyl cocrystal achieves 80% or more after 30 minutes.

It would also be interesting and beneficial to see the stability of the proposed formulation in comparison to those stored under refrigerated conditions.

- Thank you sincerely for highlighting the deficiencies in our experiments. We will actively incorporate this feedback into further research and strive to present it in additional manuscripts.

There is an absence of any statistical evaluation of the quantitative results which is necessary for publication including stating the number of replicate measurements for each sample.

- We have actively incorporated the feedback from Reviewer 3 into our manuscript, specifically in Figure 10 and 11, by including error bars, repetition trial counts, and relative standard deviations. These modifications have been highlighted in red in the manuscript. Thank you sincerely for your valuable insights.

Finally, I would urge the authors to reconsider the title as the stability and dissolution aspect are only small aspects of the research undertaken (while providing the headline results) in the manuscript and more focus should be directed to the physico-chemical characterisation of the cocrystal.

- Thank you sincerely for your valuable suggestions. We are currently planning a follow-up study, where we intend to conduct a detailed investigation into physico-chemical characterization alongside PK. Once again, we appreciate your insights.

Line 60 - Some API face challenges such as low solubility, resulting in decreased bioavailability and stability issues. This sentence is not grammatically correct and as "API" is plural here, it should be "Some APIs face challenges..."

- I sincerely appreciate the thoughtful comments from Reviewer 3. Accordingly, I have made revisions to the manuscript, replacing the term with "APIs" and highlighting the changes in red.

Line  69 In this manner, cocrystal in pharmaceutical formulations is predicted to be an effective method for enhancing solubility, bioavailability, and stability of API [7-9]. Consider rephrasing for clarity, such as "Thus, cocrystals in pharmaceutical formulations are predicted to effectively enhance the solubility, bioavailability, and stability of APIs."

- Thank you sincerely for providing the revised content. Consequently, I have made the adjustments to the manuscript as per Reviewer 3's suggestions and highlighted the changes in red. Your ongoing support is greatly appreciated.

Noltec is spelled incorrectly (Noitec) in Figure 11

- I have reviewed and addressed the mentioned points. Kindly, take a look at Figure 11 in the manuscript.

Sincerely yours,

Ji-Hun An, Ph.D.

CEO

Unicel Lab

Uiwang, Gyeonggi, 16079, Republic of Korea
